# Latent Plans for Task-Agnostic Offline Reinforcement Learning

**Erick Rosete-Beas**[1*], **Oier Mees**[1*], **Gabriel Kalweit**[1], **Joschka Boedecker**[1],**Wolfram Burgard**[2]
[1]University of Freiburg  [2]University of Technology Nuremberg

http://tacorl.cs.uni-freiburg.de

**Abstract:** Everyday tasks of long-horizon and comprising a sequence of multiple implicit subtasks still impose a major challenge in offline robot control. While a number of prior methods aimed to address this setting with variants of imitation and offline reinforcement learning, the learned behavior is typically narrow and often struggles to reach configurable long-horizon goals. As both paradigms have complementary strengths and weaknesses, we propose a novel hierarchical approach that combines the strengths of both methods to learn task-agnostic long-horizon policies from high-dimensional camera observations. Concretely, we combine a low-level policy that learns latent skills via imitation learning and a high-level policy learned from offline reinforcement learning for skill-chaining the latent behavior priors. Experiments in various simulated and real robot control tasks show that our formulation enables producing previously unseen combinations of skills to reach temporally extended goals by "stitching" together latent skills through goal chaining with an order-of-magnitude improvement in performance upon state-of-the-art baselines. We even learn one multi-task visuomotor policy for 25 distinct manipulation tasks in the real world which outperforms both imitation learning and offline reinforcement learning techniques.

**Keywords:** Offline Reinforcement Learning, Imitation Learning, Robot Learning

## 1 Introduction

In recent years, reinforcement learning (RL) has achieved tremendous successes in a variety of domains [1, 2, 3, 4]. Especially offline RL [5, 6, 7, 8, 9, 10] with its appealing property to estimate (close-to) optimal policies from previously collected and fixed datasets yielded a strong current in robot control research. However, despite the exceptional progress in this fast-moving field, current offline RL methods are often evaluated on highly specific and artificial benchmarks lacking the complexity and long-term dependencies of everyday tasks, which inherently entail a sequential relationship of multiple implicit subtasks. It lies in the nature of such composite tasks that this translates to estimating optimal actions for a significant amount of consecutive decision steps, making learning of such optimal policies very difficult. In fact, Kidambi et al. [11] discovered a quadratic relationship between the horizon of a task and the worst-case accumulated error of any offline RL method. This poses a major challenge especially in case of raw and unstructured sensory inputs, as robots must be capable of learning a large repertoire of skills and combine them to perform everyday tasks acting on long time scales.

One way to alleviate the problem of long horizons is the hierarchical subdivision of a task into high- and low-level policies, where a high-level policy is chaining executions of multiple low-level policies over primitives. Most commonly, such hierarchical structures are rather rigid and act upon a *fixed* number of low-level policies, thus lacking the flexibility and extendability required for most real-life settings. In addition, the *a priori* definition of useful low-level *skills* or their discovery from data is a highly non-trivial task. Related prior work attempted to solve this via a goal-conditioned reformulation [12] of *Conservative Q-learning* [6]. However, it was shown that on short and distinct robot manipulation tasks, self-supervised learning on unlabeled *play* can significantly surpass the performance of individual expert-trained behavioral-cloning policies [13] – which, on the flip-side, can be on par with computationally expensive offline RL methods in such settings [14, 15]. In this work, we thus propose to leverage *play data*, i.e., non-goal-directed collections of trajectories grounded

---

[*]Equal Contribution

6th Conference on Robot Learning (CoRL 2022), Auckland, New Zealand.

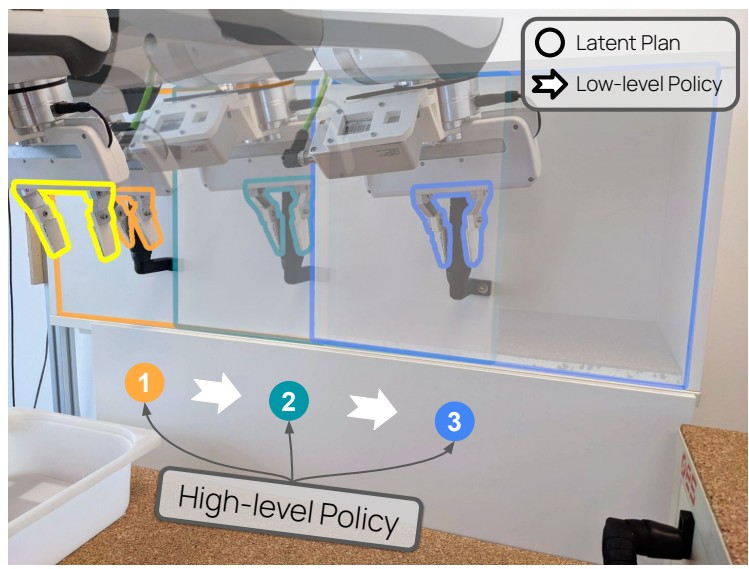

*Figure 1.* **TACO-RL** learns a single 7-DoF hierarchical visuomotor policy from offline data. It can solve long-horizon robot manipulation tasks by using a high-level policy that divides a task into a sequence of latent behaviors that are executed by a low-level policy that interacts with the environment. It reduces the effective horizon of the high-level policy and learns to chain skills through dynamic programming.

in human action execution, to estimate *short-horizon* expert policies via *imitation learning* chained via a coarse-grained high-level policy optimized by *offline RL* to account for optimal solutions over *long horizons*. By stitching latent plans extracted from unstructured data, our formulation offers the simplicity of imitation from collected play data while offering long-term optimality for sequential multi-tier tasks. Specifically, we use the collected data to learn our hierarchical policy as acquiring data with good state coverage and visual variety is important for successful applications of offline RL. Play data assumes access to an unsegmented teleoperated dataset of semantically meaningful behaviors provided by users, without a set of predefined tasks in mind. Unlike previous hierarchical approaches, that learn long-horizon tasks by performing a discrete set of tasks with a low-level policy, we are motivated by the idea of an agent capable of task-agnostic control. In this setting, we are endowing the agent with the ability to reach any possible target state from a given initial state. Specifically, we learn a low-level policy that decodes motor control actions from latent skills learned through imitation learning in a self-supervised manner. This low-level policy can perform various behaviors when present in a state by conditioning the policy with a latent plan. Consequently, we also learn a high-level policy that will order these behaviors to achieve long-horizon tasks. This high-level policy is modeled as a goal-conditioned policy that outputs latent plans to be decoded by the low-level policy. The skill-chaining of the behavior priors is learned through offline RL by augmenting plan transitions with hindsight relabeling. This hierarchical approach constitutes a practical solution by decomposing a whole task into smaller chunks of sub-tasks. The high-level policy can learn long-horizon tasks as the effective episode horizon is reduced and it does not need to capture in detail the physics of the world, simplifying the underlying dynamics of the RL agent. The model formulation allows to design a general-purpose training objective by considering every possible state reached in the data as a potential task. Additionally, our approach is effective for learning policies from large and diverse datasets which do not necessarily contain optimal behaviors.

The primary contribution of this work is an hierarchical self-supervised approach to learning task-agnostic control policies from high-dimensional observations by combining model-free RL methods with imitation learning. To our knowledge, our method is the first learning system explicitly aiming to solve long-horizon multi-tier tasks from purely offline and unstructured play data without access to a model. We integrate our components in a unified framework, called Task-AgnostiC Offline Reinforcement Learning (TACO-RL). See Figure 1 for an overview. We show that our model obtains the highest success rate when tested against other state-of-the-art baselines on various long-horizon tasks of the challenging CALVIN environment [16] and that it is able to learn a single visuomotor 7-DoF policy that can perform a wide range of long-horizon manipulation tasks in both a simulated and a real-world tabletop environment. At test time, the real world system is capable of solving a challenging suit of 25 manipulation tasks at 10 Hz that involve more than 300 decisions per task.

## 2   Related Work

**Offline Reinforcement Learning.** Offline RL [5], i.e., RL from fixed and possibly mixed transition sets, constitutes a recent trend in RL and robot control research. Generally, at least in model-free

settings, these techniques put a regularization on out-of-distribution actions, so as to enforce the learned policy to remain in the coverage of the dataset, since offline RL methods tend to suffer strongly from the problem of value overestimation. The simplest, yet competitive, attempt to solve this issue is a *behavioral cloning* addendum to a classical actor-critic framework [8]. *Conservative Q-learning* [6], on the other hand, imposes a penalty for actions not covered in the dataset making out-of-distribution actions non-optimal. *Fisher-BRC* parameterizes the critic as the log-behavior-policy [17] extended by a weighted offset term. *Implicit Q-learning* [10] modifies the Bellman optimality update towards a *SARSA*-like update, maximizing only over actions in the data-set. However, the rationale behind most current offline RL methods remains rather similar. While we make use of *Conservative Q-learning* in our experiments – which recently emerged as one of the most widely used benchmarks in offline RL – we want to point out that any other sophisticated improvement upon the classical offline RL objective is orthogonal to our work and could in principle be incorporated in our framework.

**Hierarchical policy learning.** Hierarchical policy learning involves learning a hierarchy of policies where a low-level policy performs motor control actions and a high-level policy directs the low-level policy to solve a task. While some works [18, 19, 20] learn a discrete set of lower-level policies, each behaving as a primitive skill, this is not appropriate for a general-purpose robot that accomplishes a continuum of behaviors. A large body of hierarchical policy optimization approaches following a similar rationale to our method use planning in latent state spaces [21, 22, 23, 24, 25, 26, 27, 28] and hence require a model covering the complex dynamics of multistep skills, which is an active field of research on its own. We alleviate the necessity of a model by estimating the high-level policy via model-free RL as opposed to model-based planning and thus keep the optimization over the continuous set of skills completely at training time. In contrast to a plethora of prior work [29, 30, 31, 32, 33, 34, 35], our approach acts in the offline paradigm as it yields a lot of appealing properties for learning robot control policies. En route to a skill-chaining policy able to solve long-horizon problems, our approach exploits unstructured play data to estimate the latent skill embeddings as opposed to other work that relies on expert data or predefined skills [36, 37, 38]. Whilst in principle also other latent skill representations could be considered [39, 40, 41, 42, 43], we build upon Play-LMP [13] as it has already shown great performance and robustness in this very setting. Our design choices are directed towards a general-purpose visuomotor agent that has a high zero-shot generalization even for complex long-horizon tasks, a setting more complex than in previous offline methods [13, 44]. In summary, our approach is aiming to solve temporally extended tasks without the necessity of a model or planning from entirely offline, unstructured, unlabeled and suboptimal data. This unique combination of properties thus adds a scalable and extendable optimization method to the toolbox of robot learning.

## 3 Mathematical Foundation

In this section, we introduce notation and define the problem setting. We model the interaction between and environment and a goal-conditioned policy as a goal-augmented Markov decision process $M = (S, A, p, r, G, p_0, \gamma)$ where $S$ represents the state-observation space, $A$ represents the action space, $p(s'|s, a)$ is a state-transition probability function, $r(s, a, s')$ represents the reward function, $G \subseteq S$ specifies the goal space, $p_0(s)$ is an initial state distribution, and $\gamma \in (0, 1)$ represents the discount factor. We note that the agent does not have access to the true state of the environment, but to visual observations. We learn in an offline manner by assuming to use a large, unlabeled, and undirected fixed dataset $\mathcal{D} = \{(s_1, a_1), (s_2, a_2), ..., (s_T, a_T)\}$. We then relabel this long temporal state-action stream to produce a dataset of trajectories $D = \{(\tau_i = (s_t, a_t)_{t=0}^k)_{i=1}^N$ that can be used to learn both the low-level and high-level policy without access to a model.

## 4 Offline goal-conditioned RL with TACO-RL

In this section, we elaborate on TACO-RL. Our proposed method considers the bottom-up approach; we start by training a low-level policy and we use it to provide a higher-level action space for a high-level policy that, due to this task division, is ideally facing an easier learning problem. First, we describe our unsupervised objective, which learns a continuous space of latent-conditioned behaviors $\pi_\omega(a|s_c, z)$ from $D$, where $s_c$ represents the current state. Afterward, we detail how to learn the high-level policy with offline RL by hindsight relabeling sub-trajectories with the aid of the previously learned low-level policy. This reduces our effective task horizon, making it easier to learn long-horizon tasks. Additionally, the low-level policy will predict actions close to the offline data distribution, bringing stability to the whole learning pipeline. See Figure 2 for an overview.

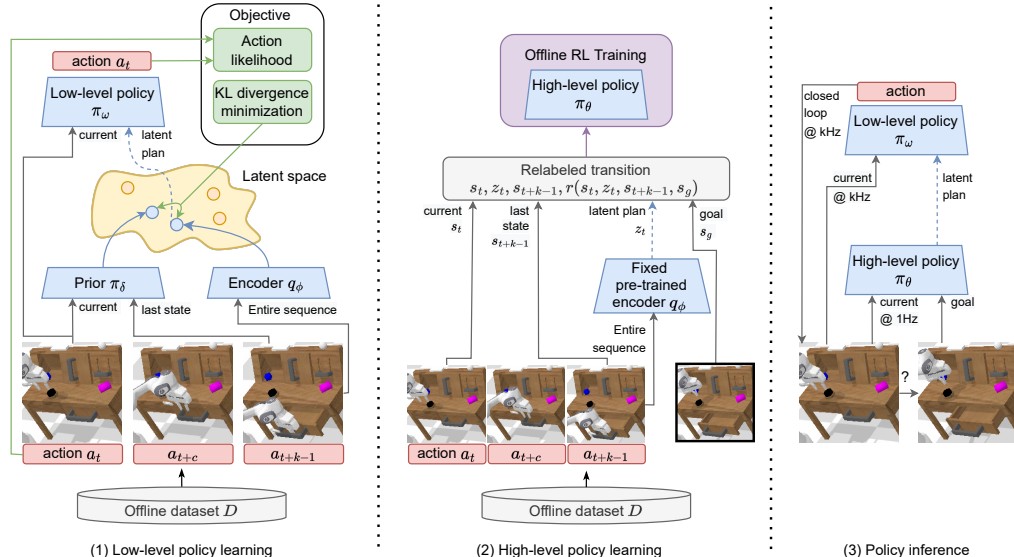

*Figure 2.* **TACO-RL Overview**. TACO-RL is a self-supervised general-purpose model learned from an offline dataset of robot interactions, it generalizes to a wide variety of long-horizon manipulation tasks. (1) Low-level policy: Recognizes and organizes a repertoire of behaviors from unlabeled, undirected dataset in a latent plan space. (2) High-level policy: Hindsight relabeling of sampled windows of experience into reward-augmented latent plan transitions. Learned with offline RL, this allows the high-level policy to stitch plans together to achieve complex long-horizon tasks. (3) Inference: the hierarchical model is used to perform goal-conditioned rollouts in robot manipulation tasks.

## 4.1 Learning the low-level policy

We would like to extract a continuous space of primitives that propose meaningful behaviors for an agent to take within a given state. We learn a low-level policy $\pi_\omega(a|s_c, z)$ from the offline, unstructured dataset $D$ that is able to decode a latent plan $z$ to its respective motor-control actions $a$. After training, we can use the latent plans as an action space for the high-level policy to learn reaching temporally extended goals by "stitching" together latent skills through goal chaining.

In our fixed static dataset $D$, it is expected to find different valid behaviors achieving the same outcome in a scene, e.g. closing a drawer quickly or slowly. We address this inherent multi-modality by auto-encoding contextual data through a latent plan space with a sequence-to-sequence conditional variational auto-encoder (seq2seq CVAE) [13, 45]. Conditioning the action decoder on the latent plan allows the policy to use the entirety of its capacity for learning uni-modal behavior. Consequently, we propose the following objective for learning the low-level policy $\pi_\omega(a|s_c, z)$:

$$\min_{\omega,\phi} \mathbb{E}_{\tau \sim D, z \sim q_\phi(z|\tau)} \left[ -\sum_{t=0}^{|\tau|} \log(\pi_\omega(a_t|s_t, z)) \right], \tag{1}$$

where $\mathbb{E}$ indicates empirical expectation and $q_\phi(z|\tau)$ may be interpreted as the latent plan encoder. As an additional component of the algorithm, we enforce consistency in the latent variables predicted by encoder $q_\phi(z|\tau)$ and prior $\pi_\delta(z|s_t, s_g)$. Since our goal is to obtain a latent plan $z$ that captures a temporal sequence of actions for a given trajectory $\tau = (s_0, a_0, ..., s_k, a_k)$, we utilize a regularization that enforces the distribution $q_\phi(z|\tau)$ to be close to just predicting the primitive or the latent variable $z$ given the initial and last state of this sub-trajectory, i.e., $\pi_\delta(z|s_c, s_g)$. The Evidence Lower Bound (ELBO) [46] for the CVAE can be written as:

$$\log p(x|s) \geq -\text{KL}\left(q(z|x, s) \parallel p(z|s)\right) + \mathbb{E}_{q(z|x,s)}\left[\log p(x|z, s)\right]. \tag{2}$$

The conditioning of the prior $\pi_\delta(z|s_c, s_g)$ on the initial and final state regularizes the distribution $q_\phi(z|\tau)$ to not overfit to the complete sub-trajectory $\tau$. In practice, rather than solving the constrained optimization directly, we implement the KL-constraint as a penalty, weighted by an appropriately chosen coefficient $\beta$. Thus, one may interpret our objective as using a sequential $\beta$-VAE [47]. Finally, we use balancing terms within the KL loss [48, 49], see Appendix D.1.

## 4.2 Offline RL with Hindsight relabeling

After distilling learned behaviors from $D$ in terms of an encoder $q_\phi(z|\tau)$, a latent behavior policy $\pi_\omega(a|s_c, z)$, and a prior $\pi_\delta(z|s_c, s_g)$, TACO-RL then applies these behaviors to learn a general-purpose agent with offline RL. We formulate a goal augmented MDP by augmenting environment trajectories with a reward function. Thereby, we sample a trajectory from the dataset $\tau =\sim D$.

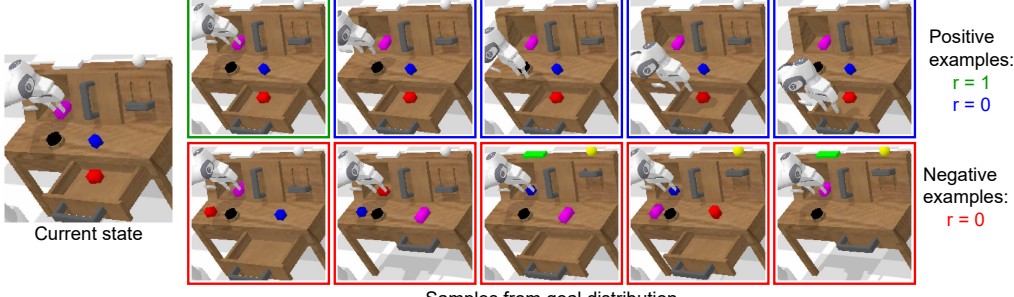

Figure 3. We relabel sampled trajectories into reward augmented transitions by sampling goal states that can be reached after executing a sequence of behaviors. With green border, we have the frame found at the end of the sampled trajectory. As this state will be reached after executing the latent behavior, the reward for this transition is 1. With blue border, we find future states that occur after the sampled sequence. These goals are necessary for chaining behaviors. The reward for these transitions is 0. Finally, with red border, we present images with similar proprioceptive information to the final state in the sampled trajectory, but a different scene arrangement. The reward for these transitions is 0.

Then, we use this trajectory to represent a high-level policy transition by using the pre-trained fixed encoder. For this, we sample a latent plan from the policy encoder $z_t \sim q_\phi(z|\tau)$ and we generate an interaction transition using the sampled latent plan $(s_t, z_t, s_{t+k-1})$. To augment this transition with a reward, we use hindsight relabeling with a sparse reward as follows $r(s_t, z_t, s_{t+k-1}, s_g) = \mathbb{1}_{s_{t+k-1}=s_g}$. In this formulation, we assume that during inference we can decode the latent plan using the low-level policy and we will reach the final trajectory state $s_{t+k-1}$. Note that $s_t$, $s_{t+k-1}$, and $s_g$ all represent images, and the reward is only given when the high-level transition reached state and goal state exactly match. During training, goals are sampled according to a distribution $s_g \in S$, which we will discuss later. Our Q-learning approach corresponds to the following Bellman error optimization objective:

$$\min_\lambda \mathbb{E}_{\tau \sim D, z \sim q_\phi(z|\tau), s_g \sim S} \left[ Q_\lambda(s_t, z_t, g) - \hat{Q}(s_t, z_t, g) \right]^2$$

$$\text{where: } \hat{Q}(s_t, z_t, g) = \left( \mathbb{1}_{s_{t+k-1}=s_g} + \gamma \mathbb{1}_{s_{t+k-1} \neq s_g} \max_{z_{t+k-1}} Q_\lambda(s_{t+k-1}, z_{t+k-1}, s_g) \right) \qquad (3)$$

We can then learn a high-level policy $\pi_\theta(z|s_c, s_g)$ with an off-the-shelf offline actor-critic method. In TACO-RL, we use Conservative Q-Learning (CQL) [6] where we initialize the actor weights with the previously learned prior policy $\pi_\delta(z|s_c, s_g)$, as the prior policy is already a good starting point that is able to solve short-horizon tasks.

**Selecting goals for relabeling transitions.** Our high-level policy is learned via offline RL as we want to learn through dynamic programming how to chain skills to reach long-horizon goals. For this we need to create a formulation that sample goal states $s_g$ that can be reached after executing a sequence of plans. Naively choosing $s_g$, say by sampling random states uniformly from the dataset, will provide an extremely sparse reward signal, as two random state images will rarely be identical. The sparse reward problem can be mitigated by selectively sampling as goals the states that were reached in future time steps along the same trajectory as $s_t$ [50]. As we want to sample states that are reached after executing a plan, we assume an arbitrary window size $k$. Concretely, to sample goals for a transition at time step $t$, we sample a discrete-time offset $\Delta \sim Geom(p)$, with $p \in [0, 1]$, and use the state at time $t + \Delta * (k - 1)$ as the goal. Note that if we assume consistent transitions from latent plan decoding, then if $\Delta = 1$, the reward for this transition is 1, as the low-level policy will reach the specified state after executing the plan, avoiding the sparsity issue.

However, relabeling all transitions in this manner introduces a problem: because the distance function is only trained on goals that have been achieved, it will systematically underestimate the distance to unreachable goals. We require a method for selecting "negative" goals that are distant, but still relevant. Randomly selecting states will produce pairs of images that are likely to be distant, but not necessarily relevant (e.g., pairs in which all objects and the robot have been moved). We want a goal sampling procedure that generates less obvious examples of distant states that are more informative. Similar to Tian et al. [51], we sample negative goal states $s_g$ which have a similar proprioceptive state. This constraint enables the Q-function to learn to focus on the scene's under-actuated parts (e.g., objects), which are likely to have distinct positions. As a result, these timesteps act as hard negatives, encouraging the model to pay closer attention to the scene. This sampling approach is computationally inexpensive, as we can query a precomputed k-nearest neighbors structure.

# 5 Experimental Results

We evaluate TACO-RL for learning a general-purpose robot in both simulated and real-world environments. The goals of these experiments are to investigate: (i) whether our hierarchical model is effective in performing complex long-horizon skills, (ii) how TACO-RL compares with alternative goal-conditioned policies, (iii) if our model scales to be used in real-world robotics.

## 5.1 Experimental Setup

We evaluate our approach in both simulated and real-world environments. We first investigate learning 7-DoF visuomotor robot skills in the CALVIN environment [16]. We train on the environment D of CALVIN, which contains 6 hours of unstructured play data collected via teleoperating a Franka Emika Panda robot arm to manipulate objects in a 3D tabletop environment.

**Baselines Methods.** As we aim to combine the complementary strengths of both paradigms, imitation and offline RL, we compare TACO-RL to representatives of the two extrema of the spectrum: the offline RL method *Conservative Q-learning* [6] extended by hindsight relabeling (CQL+HER) and the imitation learning method *Play-supervised Latent Motor Plan* (LMP) [13]. CQL+HER is trained on the derived reward as explained in Section 4.2 and introduces a penalty for out-of-distribution actions to limit the respective values of unseen actions. To make a fair comparison, this baseline is also trained with the negative mining trick. LMP, on the other hand, trains a goal conditioned imitation agent and resembles the low-level policy in TACO-RL, however, without any long-term optimality guarantees which TACO-RL accounts for by offline RL of a higher-level plan-selective policy. Additionally, we also compare against *Relay Imitation Learning* (RIL) [32]. This algorithm is the most related hierarchical algorithm, as it also learns from offline play data. RIL represents the family of methods that learns to predict latent subgoals for a low level policy.

## 5.2 Simulation Results

We start by evaluating our approach in the CALVIN environment [16]. This is a challenging environment as the scene changes through time and we act by using only RGB images of a static camera as input. As there is no predefined reward signal in this dataset, we relabel the transitions analogously as we do with TACO-RL. We investigate if our method is capable of performing complex long-horizon tasks in a robot control setting. We first attempt to solve 500 unique chains of 5 image-based goals queried in a row. For each subtask in a row the policy is conditioned on the current sub-goal image instruction and transitions to the next sub-goal only if the agent successfully completes the current task or if 180 timesteps have passed without reaching a success. We call this evaluation of performing multiple tasks on a row, long-horizon multitask with visual observations *LH-MTVis*. This setting is very challenging as it requires agents to be able to transition between different subgoals. Additionally, we ablate our model by increasing the negative goals ratio to 50% and removing the negative mining.

| Method | LH-MTVis | | | | | |
|--------|----------|----------|----------|----------|----------|----------|
| | No. Instructions in a Row (500 chains) | | | | | |
| | 1 | 2 | 3 | 4 | 5 | Avg. Len. |
| Ours | **95.4%**±2 | **82%**±7.3 | **57.8%**±15 | **32.7%**±10 | **6.9%**±3.1 | **2.7**±0.3 |
| No neg. goals | 94.9%±4.5 | 69.7%±14.3 | 31.1%±15.7 | 5.5%±4.5 | 0.9%±0.8 | 2.02±0.4 |
| 50% neg. goals | 71.8%±2.2 | 27.1%±1.8 | 6.2%±2.6 | 0.2%±0.3 | 0%±0 | 1.05±0.1 |
| RIL [32] | 70.3%±3.5 | 32.9%±7.2 | 10.5%±4.01 | 2.4±0.7 | 0.1±0.1 | 1.17±0.15 |
| LMP [13] | 91.4%±2.3 | 63.3%±3.5 | 23.1%±2.2 | 3.6%±0.9 | 0.2%±0.08 | 1.8±0.08 |
| CQL+HER | 65.5%±12.7 | 25.3%±11 | 5.6%±3.2 | 0.6%±0.2 | 0% | 0.9±0.2 |

*Table 1.* Success rates of models running 500 chains per three different random seeds, using intermediate sub-goal images.

In Table 1. we can see that TACO-RL is able to outperform all baselines. This experiment demonstrates that our agent is able to transition between different sub-goal images more naturally, enabling chaining more tasks in a row. Through our ablations, we observe that the performance of our model drops significantly when increasing the negative goal ratio to 50%. This result is to be expected as reducing the number of positive examples leads to a less informative reward indicating which are the useful behaviors to reach a transition. If we remove the negative goals from the goal distribution, the agent is still able to perform more sequential tasks than the baselines, but it underestimates the distance to the goals. This results in a decreased performance compared to our full approach.

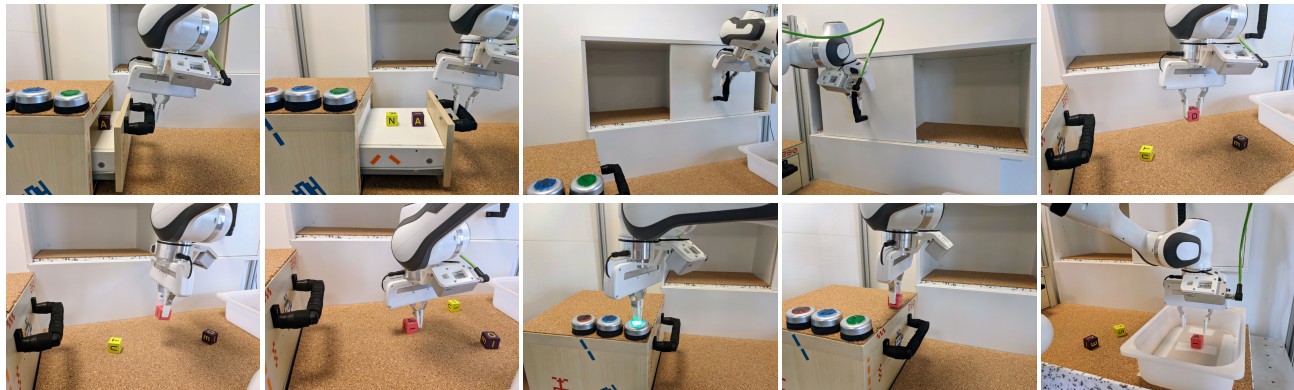

*Figure 4.* **Real-world Manipulation Tasks.** Examples shown from left to right are: closing the drawer, opening the drawer, moving the sliding door left, moving the sliding door right, lifting the block, rotating the block, pushing the block, turning the green LED on, placing the block on top of the drawer and placing the block in the container.

We then evaluate the capacity of our model to perform 1000 rollouts of two sequential tasks using a single goal image. For this, we allow the agent to perform actions until 300 timesteps has passed. We record the success rate of all models in Table 2. TACO-RL successfully performs long-horizon tasks that require reasoning over sequential behaviors with an final success rate of 27% which corresponds to an order of magnitude improvement upon the LMP and CQL+HER baselines. RIL also exploits a hierarchical structure which allows it to reason over longer horizons than the other baselines, but TACO-RL still obtains more than two times its accuracy when performing both sequential tasks using a single image, proving its effectiveness in chaining skills through dynamic programming.

We further test the capacity of the models to perform a single task when the goal image does not contain the robot performing the task, but the end effector appears in another position after the task was performed (cf. Table 3). We run 50 rollouts for each task. These harder tasks require reasoning about changes in the scene and additionally evaluates generalization, as each rollout uses a different goal image not seen during training. With our ablations, we can see that including the negative goals in our framework

| Method | LH-MTVis | | |
| --- | --- | --- | --- |
| | No. Instructions in a Row (1000 chains) | | |
| | 1 | 2 | Avg. Len. |
| Ours | **67.9%**±3.9 | **27%**±2.5 | **0.94**±0.06 |
| No neg. goals | 39.5%±2.7 | 4.2%±1.6 | 0.44±0.04 |
| 50% neg. goals | 45.4%±5.2 | 6.3%±0.6 | 0.52±0.05 |
| RIL [32] | 66.2%±6.5 | 13.3%±2.3 | 0.79±0.09 |
| LMP [13] | 34.3%±3.2 | 2.7%±0.1 | 0.3±0.03 |
| CQL+HER | 35.2%±3.4 | 2.4%±0.8 | 0.37±0.04 |

*Table 2.* Success rates of models running 1000 chains per three different random seeds conditioned only on the last goal image.

contributes greatly to obtain a good performance in this scenario alleviating to not only imitate the final end effector position, but to reach the entire scene configuration. We noticed that TACO-RL was able to outperform the baselines, which can be explained through the skill-chaining abilities of our model and the negative goal mining used to train the critic network of the high-level policy. On the other hand, CQL+HER is not able to achieve the desired goal image and LMP has a strong bias towards the end-effector position ignoring the changes in the environment. RIL can imagine intermediate latent sub-goals required to achieve the task, which reduces the bias towards the end effector position, but it stills obtain a lower accuracy rate than our method.

| Method \ Task | Place block in drawer | Open drawer | Move slider left | Turn on lightbulb |
| --- | --- | --- | --- | --- |
| Ours | **94%**±8.7 | **87.3%**±5 | **79.3%**±11.7 | **94%**±4 |
| No neg. goals | 77.3%±6.1 | 37.3%±32.3 | 13.3%±9.5 | 9.3%±6.1 |
| 50% neg. goals | 88%±7.2 | 58.7%±21.6 | 39.3%±25.3 | 92%±4 |
| RIL [32] | 74.67%±26.6 | **87.3%**±1.15 | 77%±1.4 | 92.7%±2.3 |
| LMP [13] | 78.6%±6.1 | 12%±2 | 12%±5.2 | 10%±2 |
| CQL+HER | 67.6%±20.8 | 8.6%±5 | 17.3%±8 | 4%±4 |

*Table 3.* The average success rate of goal-conditioned models running 50 rollouts where the goal image does not contain the end effector performing the task. Three models trained from different random seeds were used to perform the rollouts.

## 5.3 Real-Robot Experiments

For the real-world experiments, we investigate learning a single policy capable of performing multiple goal conditioned tasks. Examples of the tasks are shown in Figure 4. To generate the training dataset we collected nine hours of play data recorded via teleoperating a Franka Emika Panda robot arm with a VR controller. To avoid self-occlusions in the scene, these models also receive RGB

images of a gripper camera as an additional input. After training the models with the offline dataset, we performed 20 rollouts for each task using multiple goal images and start positions. TACO-RL was able to outperform the baselines consistently, especially when evaluated from a start position far away from the goal image or that required extended reasoning. We recorded the success rate of each model in Table 4.

| Task \Method | Ours | LMP [13] | CQL+HER |
|---|---|---|---|
| Lift the block on top of the drawer | **60**% | **60**% | 20% |
| Lift the block inside the drawer | **65**% | 50% | 15% |
| Lift the block from the slider | **60**% | 30% | 10% |
| Lift the block from the container | **65**% | 60% | 20% |
| Lift the block from the table | **70**% | **70**% | 30% |
| Place the block on top of the drawer | **60**% | 50% | 30% |
| Place the block inside the drawer | **70**% | 40% | 20% |
| Place the block in the slider | **30**% | 0% | 0% |
| Place the block in the container | **65**% | 30% | 15% |
| Stack the blocks | **30**% | 0% | 0% |
| Unstack the blocks | **30**% | 10% | 0% |
| Rotate block left | **70**% | 40% | 10% |
| Rotate block right | **70**% | 50% | 15% |
| Push block left | **60**% | 50% | 20% |
| Push block right | **60**% | 50% | 10% |
| Close drawer | **90**% | 70% | 20% |
| Open drawer | **70**% | 50% | 10% |
| Move slider left | **75**% | 30% | 0% |
| Move slider right | **70**% | 10% | 0% |
| Turn red light on | **60**% | 30% | 0% |
| Turn red light off | **50**% | 20% | 0% |
| Turn green light on | **70**% | 60% | 10% |
| Turn green light off | **65**% | 50% | 10% |
| Turn blue light on | **50**% | **50**% | 5% |
| Turn blue light off | **50**% | 30% | 10% |
| Average over tasks | **61**% | 40% | 11% |

*Table 4.* The average success rate of the multi-task goal-conditioned models running roll-outs in the real world.

We also tested our approach to perform sequential tasks with the real robot to verify that our approach can be scaled for long-horizon tasks. For this experiment we use a goal image for each task that the robot executes. See the supplementary video for qualitative results that showcase the diversity of tasks and the long-horizon capabilities of the different methods. Our agent trained completely from unlabeled play data is able to successfully perform most of these sequential tasks, by inferring how to transition between tasks and reach the state depicted by the goal image. More details in Appendix F.3.

Finally, we evaluate TACO-RL performing complex tasks with a single goal image, such as lifting the block and moving it to a desired position, and stacking a block on top of another. These tasks requires the agent to reason in a long-horizon manner, as if the robot imitates only the end effector position, the objects would not be arranged correctly in the scene. By stitching together the learned latent behaviors, our model was able to perform these tasks consistently.

## 6 Conclusion and Limitations

In this paper, we introduced *Task-Agnostic Offline Reinforcement Learning* (TACO-RL), which exploits a latent plan representation estimated from unstructured play data to effectively limit the horizon of a high-level offline RL policy acting upon this latent plan space. By dividing sequential multi-tier tasks into chunks of implicit subtasks solved by imitation learning, TACO-RL showed up to an order-of-magnitude improvement in performance compared to state-of-the-art both imitation learning and offline reinforcement learning baselines in both, simulated and real robot control tasks.

While TACO-RL is quite capable, it does have a number of limitations. Specifying a task to the requires providing a suitable goal image at test-time, which should be consistent with the current scene. Besides, tracking task progress might be useful when sequencing skills of different time horizons. We discuss limitations in more detail in Appendix H. But overall, we are excited about the confluence of imitation and offline RL methods towards scaling robot learning.

**Acknowledgments**

This work has been supported partly by the German Federal Ministry of Education and Research under contract 01IS18040B-OML.

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
