# OpenReview forum: "Latent Plans for Task-Agnostic Offline Reinforcement Learning"
_robot-learning.org/CoRL/2022/Conference — CoRL 2022 Poster_

### Official Review · Reviewer_mMCb · 2022-07-30

**Originality:** Good
**Technical Quality:** Very Good
**Clarity Of Presentation:** Very Good
**Impact:** 3

**Recommendation:**

Weak Accept: I recommend accepting the paper, but will not argue for my recommendation if the majority of other reviewers have a different opinion.

**Summary:**

The paper presents a method (TACO-RL), to tackle self-supervised, goal-conditioned RL from offline datasets with just visual observations. Specifically, given "play" data (unstructured human demonstrations), the method learns a policy to reach goal images, with an emphasis on reaching long horizon goals.

The proposed method learns two models:
- First, a low-level skill conditioned model trained with imitation learning. This part of the model is same as the LMP method (Lynch et al) where relabeled goals are mapped to a latent skill space which the policy is conditioned on.
- Second, a high level goal conditioned policy whose actions are the space of skills which gets fed to the low level policy. This policy is trained with offline RL with relabeled goals and a binary relabeled reward.

The proposed method outperforms each of the parts of the method (i.e. LMP and CQL+HER individually). Results look strong on a D4RL ant task, CALVIN manipulation, and a real robot setup.


**Issues:**

See Strengths and Weaknesses.

**Quality Of The Limitations Section:**

Limitations are addressed clearly

**Reviewer Expertise:**

4: The reviewer is confident but not absolutely certain that the evaluation is correct

**Robotics Focus:**

Sufficient demonstration on hardware

**Strengths And Weaknesses:**

*Strengths*
- The problem tackled is well motivated, learning to reach long-horizon goals from reward-free offline dataset is important in getting robots to learn from large datasets.
- The proposed method is straightforward and well motivated. It makes sense that the goal-conditioned imitation is reasonable on shorter horizons, but wont reach distant goals. Meanwhile, offline RL with goal-relabeling can stitch together trajectories and makes sense as a choice for a high level policy enabling reaching distant goals. Also unlike many HRL algorithms, since the low-level policy is simple imitation and decoupled from the training of the high level policy, the method seems stable and scalable.
- The simulated experiments on CALVIN show pretty solid improvements over LMP and CQL+HER individually.
- The paper has robot results.

*Weaknesses*

- First, the coverage of related work could be improved. There are several prior works [1,2,3,4,5] which also aim to reach long-horizon goal images in a self-supervised way via subgoals/hierarchy that aren't mentioned. [5] could also be good include as an additional point of comparison.

[1] Jayaraman et al. Time-Agnostic Prediction: Predicting Predictable Video Frames. 2018.
[2] Nair et al. Hierarchical Foresight. Hierarchical Foresight: Self-Supervised Learning of Long-Horizon Tasks via Visual Subgoal Generation. 2019.
[3] Pertsch et al. Keyframing the Future: Keyframe Discovery for Visual Prediction and Planning. 2019.
[4] Pertsch et al. Long-Horizon Visual Planning with Goal-Conditioned Hierarchical Predictors. 2020.
[5] Rybkin et al. Model-Based Reinforcement Learning via Latent-Space Collocation. 2021.

- Second, I'd be interested in seeing some more exhaustive ablations in the experiments. While LMP and CQL+HER are effectively the main two ablations, I'm also curious what would happen if the hard negative mining trick from Tian et al. is removed from TACO-RL. From what I can tell from the text, the baseline CQL+HER does not get this same trick, in which case the comparison isn't totally fair.

- Third, while it is great the paper has real robot results, the real world tasks look pretty short horizon, especially compared the skill chaining highlighted in the simulation results. Some longer-horizon real world tasks would make the results more compelling.

Finally, I had a few more questions about the method:
- From Figure 2 it seems like the high level policy is executed once for every 1000 steps of the low level policy. I'm curious how this parameter was chosen, and if performance varies depending on the choice of how frequently to query the high level policy.
- How important is the LMP latent plan/skill learning component of the low-level policy? Could the low-level policy be replaced with a simpler GCBC policy?

Some other minor comments:
- Table 4 caption has some formatting issues.
- It would be nice to see videos of the task execution (especially the simulated skill chaining). An anonymized website which highlights the sim and real rollouts of TACO-RL and baselines would be a nice addition to the paper.

**Summary Of Recommendation:**

Overall a solid paper which combines LMP and CQL+HER for good results on long-horizon goal reaching from offline datasets and visual observations. Method is clear, has strong improvement over baselines in sim, and has real robot results.

---

> ### Author Response · Authors · 2022-08-27
> **Response to reviewer mMCb**
>
> Thank you for the helpful feedback in your review. We address your concerns in detail below and have updated our paper accordingly.
>
> "*First, the coverage of related work could be improved. There are several prior works [1,2,3,4,5] which also aim to reach long-horizon goal images in a self-supervised way via subgoals/hierarchy that aren't mentioned. [5] could also be good include as an additional point of comparison.*"
>
> We thank reviewer mMCb for suggesting these related works, we have included them in the updated revision. We have added a hierarchical policy learning baseline, RIL, without the online finetuning. We note that the setting we are tackling in this work is quite challenging and not many works exist that try to learn a wide variety of complex robot skills from unstructured, unlabeled, offline and reset-free data and high dimensional image observations for 7-DoF continuous control in temporally extended horizons.
>
> "*Second, I'd be interested in seeing some more exhaustive ablations in the experiments. While LMP and CQL+HER are effectively the main two ablations, I'm also curious what would happen if the hard negative mining trick from Tian et al. is removed from TACO-RL. From what I can tell from the text, the baseline CQL+HER does not get this same trick, in which case the comparison isn't totally fair.*"
>
> We realize from the manuscript it might not have been clear, but the CQL+HER also uses the negative mining trick from Tian et al. We have update the manuscript to reflect this better. We have added ablation on not using the negative mining and using 50% negative goals.
>
> "*Third, while it is great the paper has real robot results, the real world tasks look pretty short horizon, especially compared the skill chaining highlighted in the simulation results. Some longer-horizon real world tasks would make the results more compelling.*"
>
> Thanks for your great suggestion. We have scaled our real world system to 25 manipulation tasks, and also evaluated sequencing several long horizon tasks of up to 300 steps, such as "move the sliding door to the right", and then "open the drawer". This is extremely challenging as the agent needs to reason how to transition between tasks. You can see the rollouts on the attached video and also on https://sites.google.com/view/taco-rl/
>
> "*From Figure 2 it seems like the high level policy is executed once for every 1000 steps of the low level policy. I'm curious how this parameter was chosen, and if performance varies depending on the choice of how frequently to query the high level policy.*"
>
> We are not using 1000 steps for our low-level policy, in fact we are using a window size of 16, we already mentioned it in the hyperparameters section of the appendix, but for completion and to avoid confusion we also added it to the main text. To clarify, TACO-RL uses the latent plans as action for the high-level policy, therefore the parameter depends on the window length of the latent plan that is sampled when learning the low-level policy. Thus, in this case the high-level is executed every 16 steps.
>
> "*How important is the LMP latent plan/skill learning component of the low-level policy? Could the low-level policy be replaced with a simpler GCBC policy?*"
>
> We thank the reviewer for this great question. Using goal conditioned behavior cloning (GCBC) as a low-level policy would imply generating subgoals, with an additional autoencoder for example, instead of latent plans.  By reasoning over continuous latent skills, our method focuses on “stitching” them to achieve temporally extended goals and combining skills seen in different episodes to produce novel behaviors. This way we efficiently try to exploit the best of the imitation learning (learn efficiently goal conditioned short horizon skills)  and reinforcement learning paradigms (long horizon reasoning over the previously acquired skills).

---

> > ### Comment · Reviewer_mMCb · 2022-08-28
> > **Re Author Response**
> >
> > Thanks for posting a detailed response and revised paper. My main concerns around fair comparison to baselines, long-horizon tasks on the real robot, and coverage of related work have all been addressed. Increasing my score to a strong accept.

---

### Official Review · Reviewer_jnvs · 2022-08-01

**Originality:** Fair
**Technical Quality:** Good
**Clarity Of Presentation:** Good
**Impact:** 3

**Recommendation:**

Weak Accept: I recommend accepting the paper, but will not argue for my recommendation if the majority of other reviewers have a different opinion.

**Summary:**

This paper presents an approach that learns a hierarchical policy from offline play data. On the low level, the propose approach imitates a latent-conditioned policy using a CVAE. On the high level, a goal-conditioned policy is trained using CQL. The proposed approach is evaluated in both simulated and real-world table manipulation tasks and achieves better performance than LMP and CQL baselines.

**Issues:**

- The design choices need to be better explained and fully justified.

- More prior work needs to be cited and discussed.

- More baselines need to be compared in the experiment section.

**Quality Of The Limitations Section:**

Additional details required

**Reviewer Expertise:**

4: The reviewer is confident but not absolutely certain that the evaluation is correct

**Robotics Focus:**

Sufficient demonstration on hardware

**Strengths And Weaknesses:**

Strengths:

- The proposed method consistently achieves a noticeable performance improvement compared to LMP and CQL+HER.

- The real-world task seems to be challenging and the proposed method achieves good success rates in these tasks.


Weaknesses:

- The paper does not clearly justify why the proposed hierarchical model is necessary. Usually, hierarchical reinforcement learning enables sharing knowledge about low-level skills across multiple tasks. The alternative and more common hierarchical RL frameworks either learn a goal-conditioned policy on the low-level and use a planner on the high level (e.g. Planning with Goal-Conditioned Policy (https://arxiv.org/abs/1911.08453), Planning-to-Practice (https://arxiv.org/pdf/2205.08129.pdf)), or learns a latent-conditioned policy on the low-level and use a non-goal-conditioned high-level policy. The proposed method is a hybrid of both and seems to be redundant to me. Could the authors explain why this design is better and include comparisons with these approaches in the experiments?

- The proposed low-level learning algorithm is reasonable but similar approaches have been proposed in prior work, e.g. Pertsch et al. (https://arxiv.org/abs/2010.11944), Fang et al. (https://arxiv.org/pdf/1910.13395.pdf), Singh at al. (https://arxiv.org/abs/2011.10024). I would recommend the authors discuss the differences with these works and include comparisons in the experiment section.

- It is not clear to me why the authors regularize q_{\phi} to a prior conditioned on both the s_c and s_g instead of only the current state. The sentence on line 143 (“Since our goal is to obtain a latent plan……” is a bit confusing to me.

- Only “three different random seeds” are used in each experiment, which is a bit insufficient.

- Is there any specific reason for using CQL instead of AWAC or IQL, which demonstrates better performance in many experiments.


- The precisions in each of Table 1-4 are not unified. Sometimes the number is rounded to the first digit sometimes the second.

- s_c is not defined in the paper.

- The limitations discussed in the paper is about goal-conditioned RL and behavioral cloning in general instead of being about the proposed method.

**Summary Of Recommendation:**

I am leaning towards rejection for now since the paper needs to better justify its design choices and include more thorough comparisons with prior work. I would consider raising my scores if the authors can address the above issues.

---

> ### Author Response · Authors · 2022-08-27
> **Response to reviewer jnvs part 1**
>
> Thank you for the insightful feedback and comments.  We address your concerns in detail below and have updated our paper accordingly.
>
> " *The paper does not clearly justify why the proposed hierarchical model is necessary. Usually, hierarchical reinforcement learning enables sharing knowledge about low-level skills across multiple tasks. The alternative and more common hierarchical RL frameworks either learn a goal-conditioned policy on the low-level and use a planner on the high level, or learns a latent-conditioned policy on the low-level and use a non-goal-conditioned high-level policy. The proposed method is a hybrid of both and seems to be redundant to me. Could the authors explain why this design is better and include comparisons with these approaches in the experiments?*"
>
> Thanks for the great question. HRL methods have traditionally struggled due to various practical challenges such as exploration, skill segmentation and reward definition. Moreover, most HRL approaches require some kind of “online” learning such as Planning-to-Practice, Relay Policy Learning or are in practice limited to short horizon and simple planar skills, such as  both aforementioned works and Planning with Goal-Conditioned Policy. However, your question points more towards why one would want to use model free RL as a high-level policy instead of a planner. There are several arguments for this: a) planning over a continuous set of skills is very costly as one has to solve a continuous maximization problem to select the best next skill/plan/goal. Second, one has to have a respective model in order to plan, which in general, is also a great limitation that our formulation does not have.
> We note that our setting  is very challenging as we learn visuomotor control policies for 7-DoF manipulators for temporally extended tasks from entirely offline, unstructured, unlabeled and suboptimal data. This setting is very scalable as it includes rich diverse behaviors and can be collected via teleop without resetting the environment or needing to segment/label the data. The key idea is to try combining the best of both paradigms of imitation learning and reinforcement learning and overcome their respective drawbacks. Thus, we learn a goal-conditioned policy with a continuous skill embedding space that can solve short-horizon tasks efficiently via imitation learning. However, imitation learning can not extrapolate or behave better than the underlying data it was trained on and moreover struggles with long-horizon goals. These points are exactly the ones which offline RL is good at and by modeling its actions as the latent plans that low-level policy can execute, we effectively reduce the task horizon and improve long-horizon reasoning. Additionally, it can “stich” together latent skills seen in different episodes to generate new behavior.
>
> "*The proposed low-level learning algorithm is reasonable but similar approaches have been proposed in prior work, e.g. Pertsch et al. Fang et al., Singh at al.. I would recommend the authors discuss the differences with these works and include comparisons in the experiment section.*"
>
> We note that our low-level learning algorithm is based on Play-LMP with few minor modifications and we compare against it in the experiments. We would like to clarify that we are not claiming to propose a novel approach for learning the low-level policy.  Our claim is that we extend a short-horizon goal conditioned latent skill based policy, trained with imitation learning, to temporally extended horizons by combining it with offline reinforcement learning as a high level policy. Unlike prior work, we learn from completely offline, diverse and unstructured data, no expert trajectories need to be collected and no environment resets need to be made, making our approach very scalable. Although Pertsch et al. also use unstructured data, they need online training to guide an SAC agent, which one might want to minimize for safety reasons in the real world, and they require scene state information for 7-DoF policy training, meaning you would need to predefine all objects in your scene and run a perception tracker to track all the objects and define/measure the rewards. Our approach can solve multiple 7-DoF manipulation tasks in the real world without the need of online exploration/finetuning or any tracking system. We have updated the paper to reflect this. CALVIN follows a similar rationale than TACO-RL, however, our work lifts the assumption of a meta-dynamics model, a highly non-trivial problem, by estimating the long-term return via offline RL. Furthermore, by estimating high- and low-level policy via offline RL, our method requires no planning during inference. Whilst the behavioral prior in PARROT simplifies exploration and bears a strong resemblance to latent plans, it does not allow to coarse the temporal resolution of some high-level policy which is the key advantage of our approach to allow for long-horizon tasks.

---

> > ### Author Response · Authors · 2022-08-27
> > **Response to reviewer jnvs part 2**
> >
> > "*It is not clear to me why the authors regularize q_{\phi} to a prior conditioned on both the s_c and s_g instead of only the current state. The sentence on line 143 (“Since our goal is to obtain a latent plan……” is a bit confusing to me.*"
> >
> > Thanks for the great question. This stems from the CVAE and the original PlayLMP formulation, during training  q_{\phi} receives a sampled window/sequence of state-action trajectory pairs (images and actions) and gets the gradient via the imitation loss. However, at test time we want to have a goal conditioned policy, so we train the prior network to match the latent distribution generated by q_{\phi}  via a KL loss, but receiving as input only the first and last frame of the sequence fed to q_{\phi}. At test time only q_{\phi} is used to generate latent plans. More details can be found in “Learning Latent Plans from Play” by Lynch et al.
> >
> > "*Only “three different random seeds” are used in each experiment, which is a bit insufficient.*"
> >
> > Due to limited time, we prioritized investing our time on implementing new baselines, doing the asked ablation studies and scaling the real world system rather than running the existing experiments with more seeds. We oriented ourselves on the D4RL paper, which we cite in Table 1 and which also reports results on three random seeds.
> >
> > "*Is there any specific reason for using CQL instead of AWAC or IQL, which demonstrates better performance in many experiments.*"
> >
> > Thanks for the great question. We noted in the manuscript that while we make use of Conservative Q-learning (CQL) in our experiments, which recently emerged as one of the most widely used benchmarks in offline RL, we want to point out that any other sophisticated improvement upon the classical offline RL objective is orthogonal to our work and could in principle be incorporated in our framework (like IQL for instance).
> >
> > "*The precisions in each of Table 1-4 are not unified. Sometimes the number is rounded to the first digit sometimes the second.*"
> >
> > Thank you for noticing this, we have updated the tables.
> >
> > "*s_c is not defined in the paper.*"
> >
> > Thanks for noticing! We have fixed this in the revised version.
> >
> > "*The limitations discussed in the paper is about goal-conditioned RL and behavioral cloning in general instead of being about the proposed method.*"
> >
> > Thank you for the feedback, we added more limitations specific to our method in the revised version.
> >
> > "*The design choices need to be better explained and fully justified.*"
> >
> > We thank the reviewer for this suggestion, we have added ablation studies that explain some of the design choices.
> >
> > "*More prior work needs to be cited and discussed.*"
> >
> > We thank the reviewer for this suggestion, we have revised our related work section in the updated manuscript.
> >
> > "*More baselines need to be compared in the experiment section.*"
> >
> > Thanks for suggestion, we have added RIL as a baseline and made several ablation experiments of our method.
> >
> > We hope this helps alleviate your issues, please let us know if you have further concerns with our submission!

---

> > > ### Comment · Reviewer_jnvs · 2022-08-28
> > > **Response to the Authors**
> > >
> > > Thank you for answering my questions. The additional paragraph in the introduction and related work section made the contribution clearer. And I appreciated the additional experiments. Given all this, I am glad to raise my score to WA.

---

### Official Review · Reviewer_3aE6 · 2022-08-04

**Originality:** Good
**Technical Quality:** Good
**Clarity Of Presentation:** Very Good
**Impact:** 4

**Recommendation:**

Weak Reject: I recommend rejecting the paper, but will not argue for my recommendation if the majority of other reviewers have a different opinion.

**Summary:**

This paper proposes a hierarchical policy and skill learning approach that can learn from human collected play data. The goal is to be able to achieve longer horizon tasks while training the hierarchical policy in a task agnostic manner. Firstly, a latent action model is learnt to encode play data. Specifically, this consists of learning a mapping sub trajectories: $s_t, s_{t+1}..., s_{t + k}$ to a latent $z_t$ which encodes the "skill". A low level policy is used to decode this latent into actions $a_t$. A policy prior is also trained to output the right $z$ to reach between state $s_t$ and goal state $s_g$.

Using the learnt encoder and policy prior, an offline RL goal-conditioned policy is learnt to produce the right $z$ given current state, achieved goal state and desired goal state. This is trained via a standard offline RL procedure and a designed goal state relabeling scheme. Experiments show good performance on CALVIN benchmark and standard offline RL benchmarks.

**Issues:**

The main issues that should be addressed:

- More comparisons to hierarchical policy learning methods
- More ablations analyzing the work
- Possible long horizon tasks, especially combining the skills seen in the real world (open drawer, grasp object etc)

**Quality Of The Limitations Section:**

Limitations are addressed clearly

**Reviewer Expertise:**

4: The reviewer is confident but not absolutely certain that the evaluation is correct

**Robotics Focus:**

Sufficient demonstration on hardware

**Strengths And Weaknesses:**

Strengths:

I think this paper tackles an important problem of learning a hierarchical policy from play data. Play data can provide useful signal and it is non-trivial to try to build a policy using this. This method, to my knowledge, is a novel way to learn a goal conditioned policy based on play data -- but the novelty could be made cleared (see weaknesses paragraph). In general, this method is technically sound and well motivated, and the presentation quality of the paper is good. Overall,  I appreciate experiments on both the offline RL benchmark, CALVIN benchmark and real world settings. In all of these, the method shows a good performance.

Weaknesses:

I think the main weakness for this paper is that it is unclear how it situates with previous work done in this area, such as LMP [13]. I think there could be more baseline comparisons, especially with other skill learning and hierarchical policy learning approaches. It is currently now quite clear how the CQL + HER baseline works - in my understanding this is just doing relabeling on the raw data. Overall the experiments could be strengthened with a more rigorous ablation section, for example understanding the role of the relabeling scheme (and comparison to others), they type of imitation learning done on the play data, which skills are actually represented by $z_t$. The real world experiments are interesting, but it would be good to see chaining of the skills for long horizon tasks -- as this is one the stated benefits of this approach. This could include for example opening the drawer, picking up the object, putting it inside the drawer and closing it.

**Summary Of Recommendation:**

My recommendation for this paper is a "weak reject" but I am willing to increase my rating if the authors can clarify exactly where this work falls with respect to prior work such as LMP and others. Overall, I think the

---

> ### Author Response · Authors · 2022-08-27
> **Response to reviewer 3aE6**
>
> We thank reviewer 3aE6 for his suggestions to improve our submission.
>
> + We have revised the related work section to help situate the approach better.
>
> + Regarding your comment about Play-LMP,  our low-level learning algorithm is based on Play-LMP with few minor modifications and we compare against it in the experiments. Our claim is that we extend a short-horizon goal conditioned latent skill based policy, trained with imitation learning, to temporally extended horizons by combining it with offline reinforcement learning as a high level policy. As Play-LMP is trained with a imitation learning loss, it can get only as good as the underlying dataset, which is a limitation of the imitation learning paradigm. The key idea of our approach is to use the latent plans learned with Play-LMP to model the actions of an offline reinforcement learning policy. This effectively reduces our effective task horizon, making it easier to learn long-horizon tasks and getting all the additional benefits of the paradigm of offline reinforcement learning, such as “stitching” latent plans from different episodes to produce new behavior and improved long-horizon reasoning.
>
> + Regarding the CQL+HER baseline, we note that it also gets the negative mining trick, making it very close to the Actionable Models Paper from Chebotar et al. We have updated the paper to reflect this.
>
> + Regarding the ablations, thanks for the suggestion, we have updated the paper with some ablation studies.
>
> + What do you mean by what type of imitation learning is done on the play data? We describe the low-level policy in section 4.1, but additional details can be found in the original paper “Learning Latent Plans from Play” by Lynch et al.
> + We have also added a hierarchical baseline RIL, as requested.
> + As you requested longer horizon skills and chaining skills in the real world, we have scaled our real world system to 25 manipulation tasks, and also evaluated sequencing several long horizon tasks of up to 300 steps, such as "move the sliding door to the right", and then "open the drawer". This is extremely challenging as the agent needs to reason how to transition between tasks. You can see the rollouts on the attached video and also on https://sites.google.com/view/taco-rl/
>
> We hope this helps alleviate your concerns, please let us know if you have further concerns with our submission!

---

> > ### Comment · Reviewer_3aE6 · 2022-08-28
> > **Re: Response to reviewer 3aE6**
> >
> > Thank you for your response. My concerns have been addressed and I will change my rating to "weak accept"

---

### Official Review · Reviewer_nPLz · 2022-08-09

**Originality:** Good
**Technical Quality:** Good
**Clarity Of Presentation:** Very Good
**Impact:** 3

**Recommendation:**

Weak Accept: I recommend accepting the paper, but will not argue for my recommendation if the majority of other reviewers have a different opinion.

**Summary:**

The authors propose an approach for learning hierarchical policies from demonstration data. TACO-RL first learns a goal-conditioned, imitation learned policy from play data, similarly to [13]. Then TACO-RL uses offline reinforcement learning to train a high-level policy to propose goals for the low level, facilitated by a goal relabeling and negative mining approach to improve performance. This is then shown on a maze and robot desk environment in real and sim to outperform LMP and CQL baselines in both short-horizon and long-horizon tasks.

**Issues:**

See weaknesses above.

**Quality Of The Limitations Section:**

Limitations are addressed clearly

**Reviewer Expertise:**

3: The reviewer is fairly confident that the evaluation is correct

**Robotics Focus:**

Sufficient demonstration on hardware

**Strengths And Weaknesses:**

he paper is generally clear, as are the figures. The approach is interesting and developing scalable policies that leverage unstructered demonstrations is important. The results show it performs well, particularly for image space, interestingly it also seems to stabilize short-horizon policies as well as long. Including the code is appreciated.

My primary concern with TACO-RL is its framing with HRL and prior work that makes it difficult to judge the novelty.
- The authors minimally discuss other HRL work and frame this approach as a “planner” which I do not believe it fits into the class of. Particularly many recent works have focused on offline HRL and goal-conditioning. See:
  - [1] Data-Efficient Hierarchical Reinforcement Learning (https://proceedings.neurips.cc/paper/2018/file/e6384711491713d29bc63fc5eeb5ba4f-Paper.pdf)
  - [2] Actionable goal-conditioned policies (https://openreview.net/pdf?id=Hye9lnCct7)
  - and many more related to these works
- The authors claim “To our knowledge, our method is the first learning system for a generalist-purpose robot that uses the benefits of hierarchical policies skill-chaining latent behaviors to reason over long-horizon tasks.” This is a significant overclaim to my understanding, based on prior work such as below (but also many others in HRL):
  - [3] Relay Policy Learning: Solving Long-Horizon Tasks via Imitation and Reinforcement Learning (https://arxiv.org/pdf/1910.11956.pdf) which chains goals
  - [4] Value function spaces (https://arxiv.org/pdf/2111.03189.pdf) which chains skills
  - [5] Neural Task Programming (https://arxiv.org/pdf/1710.01813.pdf)
  - [6] GTI (https://arxiv.org/pdf/2003.06085.pdf)
  - Even the original play paper does this to an extent.
- These approaches do skill chaining particularly for robots, the first of which proposes a very similar methodology to TACO-RL.
- The authors need to discuss and compare to works such as above, as the comparison to LMP and CQL is nice, but not informative as they are not comparable methods in this class.
- With a more HRL framing the authors should compare to some of these prior works and possibly on some standard benchmarks.

A few major questions/clarifications:
- The authors of [3] discuss issues around training a high-level policy on demonstration data, when the low-level policy may not mirror it. Is this an issue at all here?
- The authors should clarify in Section 4.1 what is the standard LMP formulation and what is theirs. And highlight any changes.
- The authors should ablate over their algorithmic choices, such as the negative mining.
- It would be useful to see long-horizon tasks in real.

There are significant typos and grammatical errors. The paper should be reviewed for clarity:
- L18- reinforcement learning (RL) achieved -> reinforcement learning (RL) has achieved
- L20- fixed datasets yield a strong current in -> rephrase
- Split up paragraph 2, its hard to get through
- “generalist-purpose robot” -> “general purpose
- Many more

Minor:
- Images of every environment would be useful
- Figure 1 could be clearer
- Define LH-VisMTLC
- It would be useful to have one video with the an overview and then rollouts from each environment.


**Summary Of Recommendation:**

The approach is reasonable and the performance is good compared to baselines of LMP and CQL. But these are not the baselines I would have expected, as this is a type of HRL planner. The framing of TACO-RL as a planner versus HRL, leads to missing these ablations and baselines, and generally discussing and framing the paper within existing literature.

After rebuttals I have raised my score to weak accept.

---

> ### Author Response · Authors · 2022-08-27
> **Response to reviewer nPLz**
>
> Thank you for your insightful comments. We address your concerns in detail below and have updated our paper accordingly.
>
> "*The authors minimally discuss other HRL work and frame this approach as a “planner” which I do not believe it fits into the class of. Particularly many recent works have focused on offline HRL and goal-conditioning*"
>
> Thanks for your great suggestions. We have updated the manuscript with an expanded section about HRL.
>
> "*The authors claim “To our knowledge, our method is the first learning system for a generalist-purpose robot that uses the benefits of hierarchical policies skill-chaining latent behaviors to reason over long-horizon tasks.” This is a significant overclaim to my understanding, based on prior work such as below (but also many others in HRL):*"
>
> Thanks, we have updated this sentence.
>
> "*The authors of [3] discuss issues around training a high-level policy on demonstration data, when the low-level policy may not mirror it. Is this an issue at all here?*"
>
> Due to our design of extracting the latent plans from an interaction window, the plans that will be predicted by our high-level policy will be consistent with the ones that can be decoded by the low-level policy. This is further enforced by the CQL optimization objective that incorporates a maximization term over plans seen in the data distribution. This is one of the advantages of our formulation against proposing latent subgoals, as proposing subgoals requires adding additional constraints to guarantee that the subgoals can be reached by the low-level policy.
>
> "*The authors should clarify in Section 4.1 what is the standard LMP formulation and what is theirs. And highlight any changes*"
>
> Thanks for the question, we kindly refer to the D.1 Section of the appendix, where the main differences from the original Play-LMP formulation to ours are explained.
>
> "*The authors should ablate over their algorithmic choices, such as the negative mining.*"
>
> That is a great point. We have added ablations to cover the cases of not using negativing mining and the case with 50% negative goals.
>
> "*It would be useful to see long-horizon tasks in real.*"
>
> We agree! We have scaled our real world system to 25 manipulation tasks, and also evaluated sequencing several long horizon tasks of up to 300 steps, such as "move the sliding door to the right", and then "open the drawer". This is extremely challenging as the agent needs to reason how to transition between tasks. You can see the rollouts on the attached video and also on https://sites.google.com/view/taco-rl/
>
> Does this address your concerns?

---

> > ### Author Response · Authors · 2022-08-27
> > **Response to reviewer nPlZ**
> >
> > Please note that the updated sentence regarding the contribution/claim of the paper has been updated in the v2 of the revised PDF.
> > We have updated the aforementioned sentence to:
> >
> > "The primary contribution of this work is an hierarchical self-supervised approach to learning task-agnostic control policies from high-dimensional observations by combining model-free RL methods with imitation learning. To our knowledge, our method is the first learning system explicitly aiming to solve long-horizon multi-tier tasks from purely offline and unstructured play data without access to a model."
> >
> > We hope this clarifies the contribution of this work and adresses your point. Please let us know if you have further concerns with our submission!

---

### Meta-Review · Area_Chair_T4rF · 2022-08-14

**Recommendation:** Accept (Poster)
**Confidence:** 5

**Metareview:**

The paper presents a framework for offline RL. The key idea is to train a high-level goal-generator trained via offline RL to guide a low-level goal-conditional policy trained via imitation learning. The method is tested on the CALVIN benchmark and standard offline RL domains, as well as a real-world robot setup.

Strengths: The reviewers agree on the importance of the problem setup and the presentation quality of the paper. They also acknowledge the value of the real-world evaluation.

Weaknesses: The main criticism lies in the technical novelty of the method and its relationship to prior works. As reviewers nPLZ, jnvs, and mMCb pointed out, there are quite a few missing reference to relevant methods. And the lack of discussion and empirical & conceptual comparison to prior works in hierarchical decision making makes it difficult to judge the novelty of the work. Reviewers 3aE6 and mMCb also pointed out that the tasks in real-world evaluation are rather short-horizon and may not demonstrate the value of the hierarchical policy framework.

Post response period: The authors have addressed most concerns raised by the reviewers. In particular, the authors extended related works discussion, included additional HRL baselines, increased number of real-world task evaluations. The scale of the real-world evaluation is in particular laudable. During the post-response period, the reviewers have all increased their ratings (3 WR->WA, 1 WA -> SA). However, some doubts in technical novelty and empirical comparisons with HRL baselines still remain. Given the author response, the updated ratings and their justifications, the AC recommends the paper to be accepted with a poster presentation.

**Best Paper Nomination:**

No

---

> ### Author Response · Authors · 2022-08-27
> **Revised PDF for rebuttal**
>
> **Comment:**
>
> We thank all reviewers for their constructive feedback and for helping us make our paper a stronger submission!
> We have uploaded a revised version of the paper with the changes highlighted in blue.
> The main issues of the reviewers which we have adressed in the revised manuscript are:
> + Section 2.2: we have improved the related work with all the suggested prior works on hierarchical reinforcement learning.
> + Section 5: we have included the requested ablations of our method without negative minining and with 50% negative goals.
> + Section 5: we have added a hierarchical reinforcement learning baseline, concretely RIL (Relay Policy Learning) without the online finetuning, as it also works with unstructured data.
> + Section 5: we have scaled our real world system to: a) learn 25 distinct manipulations tasks with a single visuomotor 7-DoF policy b) evaluate sequentially composing tasks (such as first move the sliding door to the right, then open the drawer).
> + Section 6: We have added more discussion on the limitations.
> + We have created an anonimous website to visualize the rollouts for the various environments and methods at https://sites.google.com/view/taco-rl/
>
> We hope we have addressed all your concerns and questions. Please let us know if there are any concerns preventing you from raising your score.
>
>
>
>
> **Zip File:**
>
> /attachment/183999dde793e73fc24fcd652dc7f313ebda1166.zip

---

> > ### Author Response · Authors · 2022-08-27
> > **Revised PDF v2 for rebuttal**
> >
> > **Comment:**
> >
> > We noticed that the sentence regarding the claim raised by reviewer nPLz had not been updated in the previous PDF. We have fixed this in this revised version.
> >
> > **Zip File:**
> >
> > /attachment/8ea392746d2610a5ef0127fcd2739f8392ba2705.zip